# Chemical Profiling, Toxicity and Anti-Inflammatory Activities of Essential Oils from Three Grapefruit Cultivars from KwaZulu-Natal in South Africa

**DOI:** 10.3390/molecules26113387

**Published:** 2021-06-03

**Authors:** Gugulethu Miya, Mongikazi Nyalambisa, Opeoluwa Oyedeji, Mavuto Gondwe, Adebola Oyedeji

**Affiliations:** 1Department of Chemical and Physical Science, Faculty of Natural Science, Walter Sisulu University, Mthatha 5099, South Africa; gmiya@wsu.ac.za; 2Department of Chemistry, Faculty of Science and Agriculture, University of Fort Hare, Alice 5700, South Africa; nyalambisam@yahoo.com (M.N.); ooyedeji@ufh.ac.za (O.O.); 3Department of Human Biology, Faculty of Health Science, Walter Sisulu University, Mthatha 5099, South Africa; mgondwe@wsu.ac.za

**Keywords:** *Citrus paradisi*, grapefruit, essential oil, toxicity, anti-inflammatory

## Abstract

The medicinal potential and volatile composition of different parts of three cultivars of grapefruit (*Citrus paradisi*) were evaluated for their toxicity and anti-inflammatory activities. Fresh leaf and fruit peel were separately isolated by hydrodistillation for 4 h. The essential oils were subjected to GC/GC-MS analysis for chemical profile. Toxicity of the essential oils in mice were evaluated using Lorke’s method, while an anti-inflammatory assay was performed in a rat model using egg albumin-induced oedema. The oils obtained were light yellow in colour, and odour varied from strong citrus smell to mild. Percentage yield of fresh peel oil (0.34–0.57%) was greater than the fresh leaf oil yield (0.21–0.34%). D-limonene (86.70–89.90%) was the major compound identified in the leaf oil, while β-phellandrene (90.00–91.01%) dominated the peel oil. At a dosage level of 5000 mg/kg, none of the oils showed mortality in mice. An anti-inflammatory bioassay revealed that all the oils caused a significant (*p* < 0.05–0.01) reduction in oedema size when compared to the negative control group throughout the 5 h post induction assessment period. The study reveals that the oils are non-toxic and demonstrate significant anti-inflammatory activity. Our findings suggest that the leaf and peel oils obtained from waste parts of grapefruit plants can be useful as flavouring agents, as well as anti-inflammatory agents.

## 1. Introduction

Grapefruit (*Citrus paradisi*) is a subtropical tree that belongs to the Rutaceae family and is known for its sour to semi sweet fruit [1]. It is a hybrid between sweet orange (*Citrus sinensis*) and pomelo (*Citrus maxima*). Grapefruit is commonly used in food and beverages, and in agriculture to kill bacteria and fungus [2]. Interestingly, grapefruits are available in varieties named after its colour variants: “Rose Pink”, “Ruby Red” and “White Marsh” [3]. These three varieties of grapefruit have yellow-orange skin, grow up to 5–6 m tall and have dark green leaves that are about 15 cm thick. Their flesh is acidic, but the redder varieties are the sweetest (”Ruby Red” and ”Rose Pink”). Grapefruits are cultivated in the Limpopo, Mpumalanga and KwaZulu-Natal provinces in South Africa, as they require warmer subtropical environments. They are sensitive to cold weather and can die off at freezing temperatures [4].

Essential oils are widely utilized as natural medicinal components of plants [5]. They are complex mixtures of volatile components containing several individual compounds [6]. Geographical location, climatic changes, growing conditions, seasonal variation and collection times are factors responsible for chemical composition variations of particular species [7]. Certain compounds in the oil mixtures are known to possess some biological activities that lend their traits to essential oils [8].

Grapefruit essential oils contain mixtures of volatile compounds, mostly monoterpenes and some sesquiterpenes, which are responsible for their characteristic flavour. Limonene, α-thujene, myrcene, α-terpinene and α-pinene have been reported as major compounds of these three varieties of grapefruit from Nigeria [9,10], while the Pakistani *C. paradisi* (although the cultivar was not specified) had limonene and myrcene as major components of the peel oil [11].

Different pharmacological experiments on in vitro and in vivo models have reported the ability of grapefruit to exhibit antifungal, antibacterial and antioxidant properties; these are in addition to the several therapeutic uses of the fruit juice [12,13]. Moreover, phenolic compounds isolated from grapefruit have exhibited good antioxidant activity, while the volatile aroma compounds make them excellent flavouring agents [14,15,16].

With increases in health awareness and quality of life, it has been suggested that grapefruit waste (leaf and peel) should be incorporated as an essential part of diet since it does not only provide nutritional supplements but also helps against risks of several illnesses [11]. South Africa is one of the main citrus producing countries and, according to the 2019 Citrus Market report, grapefruits production stands at 14% of the total citrus produced in the country, of which 13% are exported as fruit or as fruit juice. This activity/quantity leaves behind large amounts of leaves and peels as waste [17]. Despite the acclaimed usage of grapefruit in other countries, in South Africa, the peel and the leaf of grapefruit are still considered as waste. This has also led to little or no attention being given to their therapeutic potential, aromatic chemical profiles, phytochemical constituents and their other potential uses in folklore medicine. Furthermore, there is little or no information on the various species found in South Africa to scientifically validate the medicinal use(s) of this plant. The safety profile of the grapefruits which are commonly used by communities have also not being reported hence the results obtained in this study will provide additional data on the phytochemistry, safety profile and biological activities of these valued “wastes”. This study therefore aims at determining the chemical composition of essential oils made from grapefruit leaf and peel, and evaluating their biological properties (anti-inflammatory and toxicity) in regard to their economic and medicinal value chains.

## 2. Results

Pale yellow to colourless essential oils with strong citrus odour were obtained from the fresh leaf and peel of the three grapefruit varieties studied, with percentage yield (*w*/*w*). The fresh leaf oils had percentage yields (0.34–0.57%), while the peel oils had (0.21–0.34%) (Table 1). Table 2 shows the chemical compositions of the essential oils isolated from the fresh leaf and peel of the three grapefruit varieties. The GC-MS result analysis of the grapefruit fresh leaf and peel oils showed that they contained a variety of compounds. The main compounds from the peel oils of the three varieties studied were identified as d-limonene (87–90%), β-myrcene (2–4%) and γ-terpinene (0.05–2%). By contrast, β-phellandrene (90–91%), furanoid (0.6–2%) and caryophyllene (0.08–2%) were major compounds identified in the leaf oils. The predominance of d-limonene in the peel oils indicates that these waste peels can be used medicinally for the management of gallstones, as an appetite suppressant and in foods as flavouring agents [18], while the presence of β-phellandrene as the major compound in leaf oil of all grapefruit varieties shows the potentials of the oils as antimicrobials and antifungals [19].

*Citrus paradisi* peel and leaf essential oils from the three varieties studied here showed no mortality orally at dosage levels of 5000 mg/kg (Table 3). FRPG had the highest significance, followed by FMPG and lastly FSPG (50–200 mg/kg). All the peel essential oils effectively reduced the egg albumin-induced paw oedema (*p* < 0.05–0.01) throughout the 5 h period. Although, in the negative group, oils reduced the egg albumin-induced paw oedema, their efficacy was lower than the efficacy visible in the positive control group (diclofenac), which signifies anti-inflammatory qualities. Moreover, diclofenac (a known standard drug) also reduced the egg albumin-induced paw oedema (*p* < 0.05–0.01) throughout the 5 h assessment period (Figure 1, Figure 2, Figure 3, Figure 4, Figure 5 and Figure 6). From the result of the bioassay, fresh ”Rose” peel and leaves were most active as anti-inflammatory agents, with the highest efficacy within the first two hours. Figure 7 gives a pictorial view of the cultivar that was most active during the anti-inflammatory bioassay.

## 3. Discussion

The extracted oils had a pleasant citrus aroma, which supports their use in cosmetic and pharmaceutical industries as flavouring agents. Limonene, the major constituent of the peel oil of the *C. paradisi* plant growing in Pakistan [11] and Nigeria [10], was also found as a major constituent of the South African variety used in our study. This implies that the different *C. paradisi* varieties have the same chemotype. Since there was no literature report on leaf oil composition, we are reporting that β-phellandrene was the dominating compound identified in the leaf oils of three varieties of *C. paradisi* from South Africa. Our study also reveals the anti-inflammatory potential of the oils, with the “Rose Pink” peel oils having the highest and fastest efficacy compared to the other two varieties. Results indicate that “Rose Pink” peel oils are able to reduce inflammation within the first two hours, while “Ruby Red” and “White Marsh” are only effective after the 3rd and 4th h.

This study also established that grapefruit oils are non-toxic when taken orally and can be commercialized as oral dose drugs or intravenous ones. Side effects associated with grapefruit consumption and interactions with other drugs have been reported [20]. However, a detailed toxicity study should be conducted by checking organs for the level of toxicity of the oils. Thus, further research is necessary to understand interactions with other drugs when grapefruit peel and leaf oils are taken orally at amounts considered to be safe for administration.

Our study indicates that all grapefruit essential oils can be used as anti-inflammatory agents, with Rose grapefruit showing the highest potential, followed by Marsh and, lastly, Star Ruby, in both peels and leaves. This could be linked to the known antioxidant potential of the fruit. The presence of Vitamin C, B, A and lycopene in grapefruits has been reported to help fight inflammation caused by free-radical damage in the body [21].

## 4. Materials and Methods

### 4.1. Plant Material

Three varieties of *Citrus paradisi* fruits and leaves were collected from Eshowe Mystic Farm in the KwaZulu-Natal Province, South Africa, at the beginning of the ripening period (March 2017), and the leaves and fruits were identified by Dr Immelmann of the Department of Biological and Environmental Sciences, Walter Sisulu University, Mthatha, with voucher numbers MGM 003 for Marsh, MGM 004 for Rose and MGM 005 for Star Ruby grapefruits.

### 4.2. Equipment

Weighing balance (Mettler Toledo), animal cages, Plexiglas cage, caliper, Clevenger-type distillation unit.

Reagents and drugs: Diclofenac potassium (Diclogesic Supreme), aspirin. All chemicals and reagents were obtained from Shalom Laboratory Suppliers, an outlet for Sigma–Aldrich Chemical Co. (St Louis, MO, USA). All the chemicals used, including the solvents, were of analytical grade.

### 4.3. Isolation of Essential Oils

Fresh leaves and fruit peels of *Citrus paradisi* (350 g each) were hydrodistilled using the Clevenger apparatus for 4 h to obtain essential oils [17]. Plant parts were distilled immediately after harvesting, while the dried plant parts were obtained by air-drying the fresh plant parts at room temperature over a period of two weeks. The essential oils were placed in closed vials and stored in a cool dry place [22].

### 4.4. Gas Chromatography-Mass Spectrometry

GC-MS analyses of the oils were performed on a Bruker 450 Gas Chromatograph connected to a 300 MS/MS mass spectrometer system operating in EI mode at 70 eV. It was equipped with an HP-5 MS fused silica capillary system with 5% phenylmethylsiloxane stationary phase. The capillary column parameter was 30 m by 0.25 mm, with film thickness of 0.25 µm. The initial temperature of the column was 50 °C and was heated to 240 °C at a rate of 5 °C/min; the final temperature was kept at 450 °C for a run time of 66.25 min. Helium was used as the carrier gas at a flow rate of 1.0 ml/min. The split ratio was 100:1. Scan time was 78 min with a scanning range of 35 to 450 amu. One microliter (1 µL) of the diluted oil (in hexane) was injected for analysis. N-Alkane of C8 to C30 was run using the same Kovats indices. The constituents were identified using GC retention indices obtained in our group library, along with those in the literature [23,24,25]. The retention indices were determined in relation to a homologous series of alkanes under the same operating conditions.

### 4.5. Animals

Mice and rats were obtained from the South African Vaccine Initiative, Johannesburg, and kept at the Animal Holding Facility, Department of Human Biology, Walter Sisulu University. Wistar rats (200–300 g/BW), all males, were randomly selected (n = 6), and were used for the anti-inflammatory test. Swiss mice (25–35 g/BW; *n* = 6) (g/BW is the mass of the animal unit (in grams per body weight)), all males, were used for the acute toxicity test. The animals were fed with standard laboratory food for rodents and water was provided freely except during the experiment. All the animal experiments were carried out in strict compliance with the “principle of laboratory animal care” guidelines and procedures [26].

### 4.6. Acute Toxicity Test

The essential oils obtained from the peel of the three varieties of grapefruits studied were used for the biological evaluation because of their higher yield and the presence of more compounds compared to the leaf oil (from the GC/MS analysis). The peel oil was individually emulsified with Tween 80 to obtain a maximum concentration of 5% *v*/*v* before administration to the animals. The acute toxicity of the oil was evaluated according to the modified Lorke’s method [27]. This was accomplished in 2 phases. The first phase consisted of three sub-groups with 3 animals each for the dose levels of 10, 100 and 1000 mg/kg, and a negative control group. The second phase involved 4 subgroups with 1 animal each per dose levels of 1000, 1600, 2900 and 5000 mg/kg, respectively, and a negative control group. Immediately after the treatment, each mouse was placed inside the Plexiglas cage and observed for immediate effects up to 30 min and thereafter for 24 h for lethal effects culminating in death. The LD_50_ of the essential oils were estimated as the geometric mean of the lowest dose causing death and the highest dose causing no death.

### 4.7. Anti-Inflammatory Test

Wistar rats were used for the acute inflammation study using fresh egg albumin as the phlogistic agent. Acute inflammation was studied for 5 h. Animals were divided into eleven (11) experimental groups of six rats each [28], and treated as listed below:Group I—5% Tween 80 negative control group (10 mL/kg);Group II—Diclofenac positive control (Sigma Aldrich, USA) (100 mg/kg);Group III, IV, V—fresh ‘Rose Pink’ variant peel essential oil (FRPG)—(50, 100, 200 mg/kg);Group VI, VII, VIII—fresh ‘White Marsh’ variant peel essential oil (FMPG)—(50, 100, 200 mg/kg);Group IX, X, XI—fresh ‘Ruby Red’ variant peel essential oil (FSPG)—(50, 100, 200 mg/kg).

Animals were fasted and deprived of water for 18 h before the experiment in order to ensure uniform hydration and to reduce variability in response to oedema. A 5% Tween 80, Diclofenac, FRPG, FMPG and FSPG were orally administered to the groups. One hour post treatment, inflammation at the left hind paw was induced by injecting 0.1 mL of fresh egg albumin into the sub-plantar surface of the left hind paw in all rats. The fresh egg albumin was diluted in distilled water in 1:1 ratio. The paw diameters of the animals were measured with a digital caliper. Repeated weighing was carried out. Baseline paw diameters were recorded. Hourly readings were taken after the egg albumin was introduced up to 5 h.

### 4.8. Data Presentation and Statistical Analysis

Data were presented in the form of tables and graphs. Results were presented as mean ± SEM. For anti-inflammatory analysis, one-way analysis of variance (ANOVA) was used to analyse and compare the data, followed by Dunnett’s test for multiple comparisons among groups. *p* < 0.05 was taken as the limit of significance in all cases. Statistical analyses were made using GraphPad Prism version 5.00 for Windows (GraphPad Software, La Jolla, CA, USA).

## 5. Conclusions

The leaf and peel oils from the three cultivars of *Citrus paradisi* studied have similar prominent compounds, although with slight variations in their percent composition. Biologically, these oils showed no mortality in mice during the toxicity study and demonstrated that they possess anti-inflammatory qualities. These perceived waste products have great potential and could be further studied for anti-inflammatory drug development, thus supporting the traditional application of this fruit for treating various illnesses, including inflammation.

## Figures and Tables

**Figure 1 molecules-26-03387-f001:**
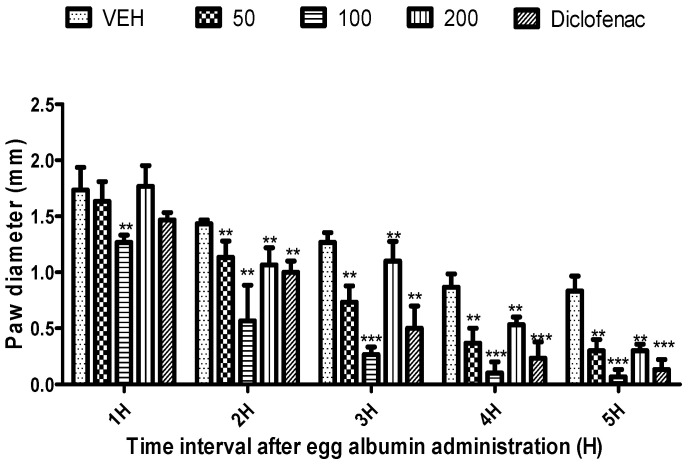
Effect of “White Marsh” grapefruit (*C. paradisi*) fresh peel essential oil on egg albumin-induced rat paw oedema. Results are expressed as Mean ± SEM. VEH, ECH and ASA represent vehicle (5% Tween 80), Marsh peel essential oil (FMPG) and diclofenac, respectively. ** *p* < 0.01, *** *p* < 0.001, statistically different from negative control group (ANOVA, Dunnett’s).

**Figure 2 molecules-26-03387-f002:**
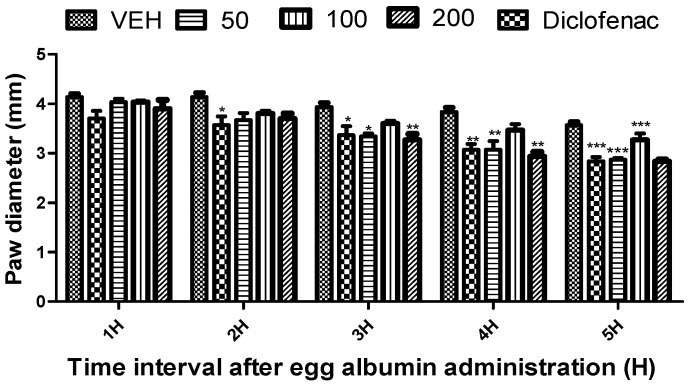
Effect of “White Marsh” grapefruit (*C. paradisi*) fresh leaf essential oil on egg albumin-induced rat paw oedema. Results are expressed as Mean ± SEM. VEH, ECH and ASA represent vehicle (5% Tween 80), Marsh leaves essential oil (FMLG) and diclofenac, respectively. * *p* < 0.05, ** *p* < 0.01 and *** *p* < 0.001, statistically different from negative control group (ANOVA, Dunnett’s).

**Figure 3 molecules-26-03387-f003:**
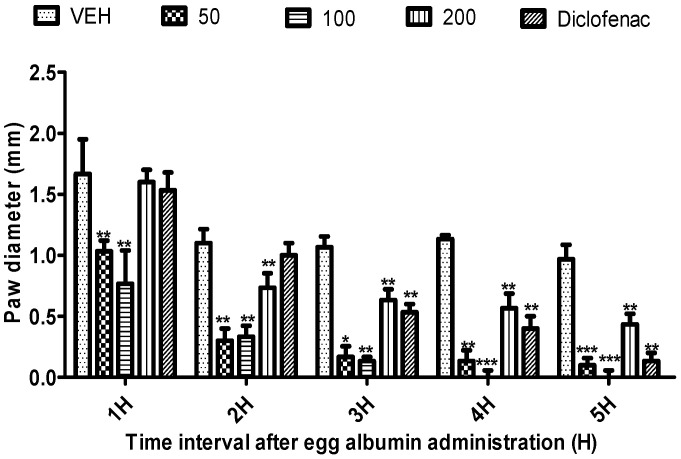
Effect of “Rose Pink” grapefruit (*C. paradisi*) fresh peel essential oil on egg albumin-induced rat paw oedema. Results are expressed as Mean ± SEM. VEH, ECH and ASA represent vehicle (5% Tween 80), Rose peel essential oil (FRPG) and diclofenac, respectively. * *p* < 0.05, ** *p* < 0.01 and *** *p* < 0.001, statistically different from negative control group (ANOVA, Dunnett’s).

**Figure 4 molecules-26-03387-f004:**
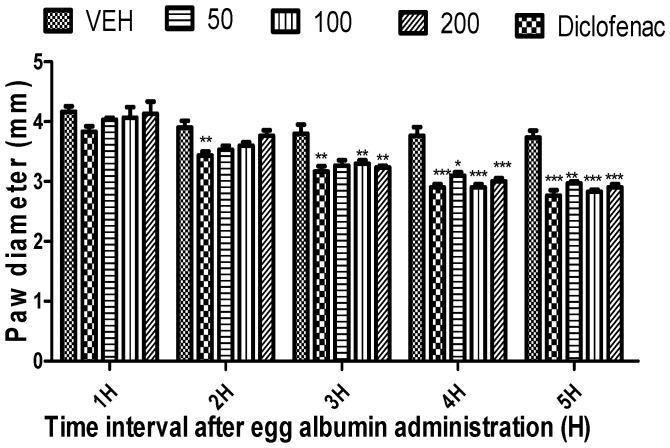
Effect of “Rose Pink” grapefruit (*C. paradisi*) fresh leaves essential oil on egg albumin-induced rat paw oedema. Results are expressed as Mean ± SEM. VEH, ECH and ASA represent vehicle (5% Tween 80), Rose leaves essential oil (FRLG) and diclofenac, respectively. * *p* < 0.05, ** *p* < 0.01 and *** *p* < 0.001, statistically different from negative control group (ANOVA, Dunnett’s).

**Figure 5 molecules-26-03387-f005:**
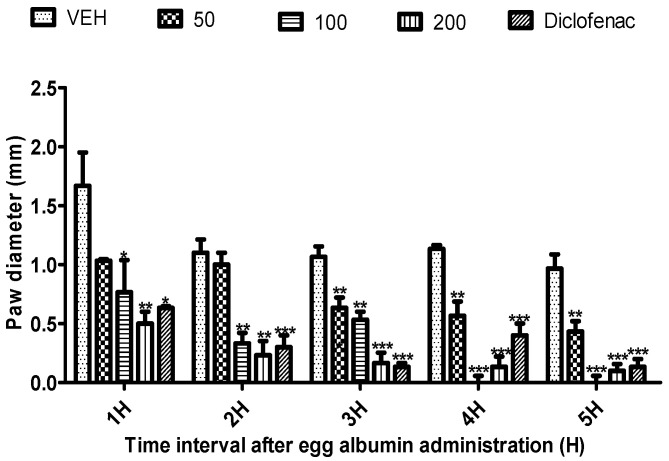
Effect of “Ruby Red” grapefruit (*C. paradisi*) fresh peel essential oil on egg albumin-induced rat paw oedema. Results are expressed as Mean ± SEM. VEH, ECH and ASA represent vehicle (5% Tween 80), Ruby peel essential oil (FSPG) and diclofenac, respectively. * *p* < 0.05, ** *p* < 0.01 and *** *p* < 0.001, statistically different from negative control group (ANOVA, Dunnett’s).

**Figure 6 molecules-26-03387-f006:**
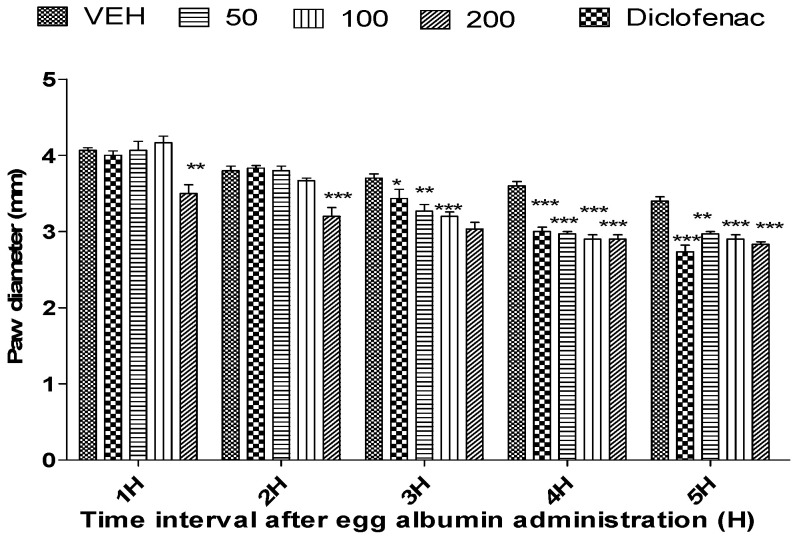
Effect of “Ruby Red” grapefruit (*C. paradisi*) fresh leaves essential oil on egg albumin-induced rat paw oedema. Results are expressed as Mean ± SEM. VEH, ECH and ASA represent vehicle (5% Tween 80), Ruby leaves essential oil (FSLG) and diclofenac, respectively. * *p* < 0.05, ** *p* < 0.01 and *** *p* < 0.001, statistically different from negative control group (ANOVA, Dunnett’s).

**Figure 7 molecules-26-03387-f007:**
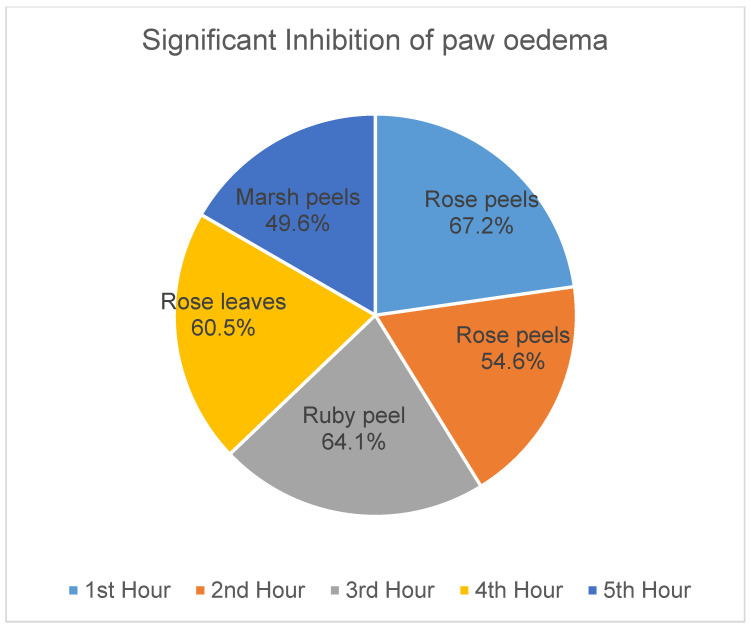
Summary of most active anti-inflammatory cultivar per hour.

**Table 1 molecules-26-03387-t001:** Physiochemical analysis of the essential oils from three *Citrus paradisi* varieties.

Plant Name and Plant Part	Smell of the Oil	Colour of the Oil	Starting Material/g	Mass of Oil/g	% (*w*/*w*)
“Ruby Red” grape fresh leaf	Mild sweet citrus smell	Colourless	318.6	1.4	0.44
“Ruby Red” grape fresh peel	Sweet citrus smell	Pale yellow	885.6	3.0	0.34
“White Marsh” grape fresh leaf	Strong lemonade smell	Colourless	386.2	0.8	0.21
“White Marsh” grape fresh peel	Sweet lemonade smell	Colourless	904.6	3.1	0.34
“Rose Pink” grape fresh leaf	Mild citrus small	Colourless	405.2	2.3	0.57
“Rose Pink” grape fresh peel	Mild citrus smell	Colourless	612.1	1.9	0.31

**Table 2 molecules-26-03387-t002:** Chemical composition of the essential oils from three *Citrus paradisi* varieties.

Compound Name	RT	KI exp	KI lit	“Ruby Red”		“Rose Pink”		“White Marsh”	
				Percentage composition of the total essential oil					
				Peel	Leaf	Peel	Leaf	Peel	Leaf
toluene	4.77	700	770	0.08	-	-	t	-	0.06
hexanal	5.10	715	780			-	1.23	-	0.98
pentylcyclopropane	7.02	800	813	0.68	-	-	-	-	-
(*Z*)-3-hexen-1-ol	8.54	600	827	0.08	-	0.10	-	0.29	-
heptanal	9.78	840	896	t	-	-	-	-	-
α-pinene	10.66	1000	933	0.26	t	0.52	-	0.64	-
β-pinene	10.86	1001	981	0.10	0.43	0.07	t	0.07	t
β-myrcene	11.23	1003	992	2.43	0.90	2.50	0.03	3.57	0.03
octanal	11.77	800	987	1.48	-	0.80	-	0.50	-
5-hepten-2-one,6 methyl	12.30	801	994	t	-	-	-	-	-
3-carene	12.96	1017	1011	t	t	-	-	-	-
β-phellandrene	13.54	1022	1031	0.66	91.01	0.60	90.00	0.60	90.05
d-limonene	13.76	1024	1034	86.70	1.21	89.90	0.03	88.06	0.01
β-ocimene	14.98	1038	1050	0.23	2.01	0.05	0.71	0.01	0.71
γ-terpinene	15.86	1049	1062	0.04	0.11	1.90	-	0.56	-
furanoid	17.10	1070	1088	0.70	-	0.05	1.98	0.05	0.62
linalool	17.76	1083	1098	1.80	0.02	0.03	0.01	-	0.01
nonanal	18.29	981	1104	0.10	-	0.02	-	0.02	-
*cis*-limonene oxide	18.42	1098	1135	0.02	-	-	-	0.01	0.73
limonene oxide	18.53	1101	1138	0.01	-	0.02	-	0.02	-
citronellal	18.84	1109	1155	0.02	0.43	0.16	-	0.16	-
terpinen-4-ol	19.12	1118	1182	0.24	0.76	0.32	-	0.10	0.21
decanal	19.23	1121	1185	0.50	-	-	-	0.12	-
α-terpineol	19.69	1137	1192	0.42	-	0.03		0.03	-
citronellol	20.27	1159	1211			0.19	0.01	0.19	0.01
*trans*-carveol	21.02	1196	1235	0.06	-	0.10	-	0.10	-
citral	21.28	1211	1240	0.38	0.03	0.13	-	0.13	-
geraniol	25.70	1358	1245	0.43	-	0.11	-	0.01	-
1-decanol	25.86	1396	1263	0.38	-	-	-	0.38	-
*p*-mentha-1(7),8(10)-dien-9-ol	26.22	1413	1287	0.01	-	-	-	0.01	-
decane,1-ethenyloxy	26.30	1200	1297	0.08	-	-	-	0.01	0.01
*p*-metha-1,8-dien-7-ol	27.18	1204	1330	0.02	-	-	-	0.02	-
α-terpinyl acetate	27.51	1223	1334	0.04	-	-	-	0.04	-
β-elemene	30.02	1283	1339	0.06	-	0.07	-	0.06	0.08
caryophyllene	30.49	1293	1445	0.51	0.62	0.08	2.25	0.43	1.30
humulene	30.57	1312	1456	0.06	-	-	-	0.06	-
β-muurolene	30.62	1326	1477	0.17	-	0.48	-	0.17	-
germacrene D	30.71	1347	1480	0.07	-	-	-	0.07	-
geranyl acetate	31.56	1368	1560	0.19	-	0.03	-	0.12	-
*trans*-nerolidol	31.95	1387	1564	0.03	0.05	-	-	0.04	0.05
copaene oxide	32.56	1409	1585	0.17	-	-	-	0.70	-
tau-muurolol	34.19	1495	1640	0.01	-	-	-	-	-
*cis*-β-fernesene	36.84	1576	1696	0.02	-	-	-	-	-
*cis*-α-bisabolene	36.84	1589	1783	-	-	0.15	-	-	-
dodacanal	39.57	1837	2217	-	-	0.06	-	-	-
TOTAL %				99.56	97.60	99.48	96.50	97.78	92.77

KI exp = (experimental Kovats indices), KI lit = (literature Kovats indices) and t = trace amount less than 0.01. KI exp = ((Rti-Rtzi/Rtz-Rti) + Cn) × 100: Rti is the retention time of the compound, Rtzi is the retention time of the lower alkane, Rtz is the retention time of the higher alkane and Cn is the number of carbons in the compound.

**Table 3 molecules-26-03387-t003:** Acute toxicity profile of the leaf and peel essential oils from three *Citrus paradisi* cultivars in mice.

“White Marsh” Grapefruit			“Rose Pink” Grapefruit		“Ruby Red” Grapefruit	
Treatment mg/kg, p.o.	Leaf	Peels	Leaf	Peels	Leaf	Peels
	Mortality after 24 h	Mortality after 24 h	Mortality after 24 h	Mortality after 24 h	Mortality after 24 h	Mortality after 24 h
**Phase 1**						
10	0/3	0/3	0/3	0/3	0/3	0/3
100	0/3	0/3	0/3	0/3	0/3	0/3
1000	0/3	0/3	0/3	0/3	0/3	0/3
**Phase 2**						
1000	0/1	0/1	0/1	0/1	0/1	0/1
1600	0/1	0/1	0/1	0/1	0/1	0/1
2600	0/1	0/1	0/1	0/1	0/1	0/1
LD_50_	≥2600 mg/kg	≥2600 mg/kg	≥2600 mg/kg	≥2600 mg/kg	≥2600 mg/kg	≥2600 mg/kg

Key note: 0/3 means no death in the three mice used in the experiment, while 0/1 means the same when only one animal was used. Phase 1 is for low dose levels, while phase 2 is for higher dose levels.

## Data Availability

Data available in a publicly accessible repository that does not issue DOIs http://vital.seals.ac.za:8080/vital/access/manager/Index?site_name=Walter%20Sisulu%20University.

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
