# Peer review of "Chemical Profiling, Toxicity and Anti-Inflammatory Activities of Essential Oils from Three Grapefruit Cultivars from KwaZulu-Natal in South Africa"

_molecules, 2021, doi:10.3390/molecules26113387_

Round 1
Reviewer 1 Report
Please see the attachment for details.

Reviewer 2 Report
The article sent for evaluation brings new content to the current state of knowledge. Provides new information on the chemical diversity of essential oils resulting from inter-varietal variation. It is interesting research showing the practical potential and possibilities of using the compared oils. The healing potential of fresh leaves and peel of Citrus paradisi fruit was comprehensively assessed.
ABSTRACT
The abstract is well composed, it contains the most important issues of the experiments carried out.
ITRODUCTION
The introduction is comprehensive and relates to the existing knowledge of the subject. However, the authors did not provide information distinguishing their study from the available reports and did not indicate the novelty of this study. Therefore, this should be completed.
RESULTS
The results are described correctly and all the results of the experiment are clearly presented.
SUMMARY
The summary contains the most important achievements of the research and tips on the possible use of grapefruit oils. The bak is an indication as to whether the variety determines the chemical composition of the volatile substances. The authors are asked to complete this information. The authors do not address interdominal differences. After all, the title of the article points to a study of three grapefruit varieties. This aspect should be taken into account in the description of the results. Not only describing the obtained oils and their amount, but also the toxic and anti-inflammatory effects of the bio-test. The authors are asked to indicate which of the analyzed varieties is more useful. As they had a different composition of essential oil components, this should be particularly emphasized.
MATERIALS AND METHODS
The analytical procedure for sample preparation and analysis is described. However, to make them clearer and more accurate, I suggest dividing this chapter into subsections:
Preparation of the raw material: which should indicate how the samples were handled. It is not known if they were shredded? Or maybe they weren't? What was the size of the shredded material?
Essential oil distillation: where all details of this procedure should be included. How much water was used for distillation. After all, grapefruit does not have a detailed monograph, especially with regard to leaves.
GC / MS analysis: which must contain all information about the apparatus, including the factory name of the chromatograph.
And also how the content of volatile components was calculated.
Reagents: Please specify the angle obtained and where you purchased standards and calibers, solvents etc.
The same applies to the analysis of the toxicity of oils. Please describe Lorke's method in detail. The exact number of subjects on which the toxicity study was conducted should also be indicated. It is essential in this type of research.
Also for the rat anti-inflammatory bioassay.
REFERENCES
Please refresh the literature a bit. It's a bit dated nowadays. In the available world literature there are many references that can be cited, and they are from the last 10 years.
The entire article should be carefully examined so that it is edited according to the journal's recommendations. Unfortunately, there are many editorial errors in the text at present. This needs to be corrected. Among others: Citrus paradisi, to be written in italics. And the names of the varieties in this way 'Rose'.
Round 2
Reviewer 1 Report
Now the manuscript looks better. However, I have following comments:
- Line 139, This could to linked to the known antioxidant potential of the fruit. Please check the sentence meaning.
- Line 166, Citrus paradisi (350g each). Use a space before ‘g’.
3. Hour is written as ‘h’ and ‘H’. Please use one of them in the whole manuscript, not both. Also insert a space before the unit.

Reviewer 2 Report
The authors significantly improved the manuscript as recommended by the reviewer. Doubts have been resolved. The manuscript in its current form may be published.